# Decisions, Counterfactual Explanations and Strategic Behavior

**Stratis Tsirtsis**
Max Planck Institute for Software Systems
Kaiserslautern, Germany
stsirtsis@mpi-sws.org

**Manuel Gomez-Rodriguez**
Max Planck Institute for Software Systems
Kaiserslautern, Germany
manuelgr@mpi-sws.org

## Abstract

As data-driven predictive models are increasingly used to inform decisions, it has been argued that decision makers should provide explanations that help individuals understand what would have to change for these decisions to be beneficial ones. However, there has been little discussion on the possibility that individuals may use the above *counterfactual explanations* to invest effort strategically and maximize their chances of receiving a beneficial decision. In this paper, our goal is to find policies and counterfactual explanations that are optimal in terms of utility in such a strategic setting. We first show that, given a pre-defined policy, the problem of finding the optimal set of counterfactual explanations is NP-hard. Then, we show that the corresponding objective is nondecreasing and satisfies submodularity and this allows a standard greedy algorithm to enjoy approximation guarantees. In addition, we further show that the problem of jointly finding both the optimal policy and set of counterfactual explanations reduces to maximizing a non-monotone submodular function. As a result, we can use a recent randomized algorithm to solve the problem, which also offers approximation guarantees. Finally, we demonstrate that, by incorporating a matroid constraint into the problem formulation, we can increase the diversity of the optimal set of counterfactual explanations and incentivize individuals across the whole spectrum of the population to self improve. Experiments on synthetic and real lending and credit card data illustrate our theoretical findings and show that the counterfactual explanations and decision policies found by our algorithms achieve higher utility than several competitive baselines.

## 1 Introduction

Whenever a bank decides to offer a loan to a customer, a university decides to admit a prospective student, or a company decides to hire a new employee, the decision is increasingly informed by a data-driven predictive model. In all these high-stakes applications, the goal of the predictive model is to provide accurate predictions of the outcomes from a set of observable features while the goal of the decision maker is to take decisions that maximize a given utility function. For example, in university admissions, the predictive model may estimate the ability of each prospective student to successfully complete the graduate program while the decision maker may weigh the model's estimate against other socio-economic considerations (*e.g.*, number of available scholarships, diversity commitments).

In this context, there has been a tremendous excitement on the potential of data-driven predictive models to enhance decision making in high-stakes applications. However, there has also been a heated debate about their lack of transparency and explainability [1–5]. As a result, there already exists a legal requirement to grant individuals who are subject to (semi)-automated decision making the *right-to-explanation* in the European Union [6, 7]. With this motivation, there has been a flurry of work on interpretable machine learning [8–16], which has predominantly focused on developing methods to find explanations for the predictions made by a predictive model. Within this line of work,

the work most closely related to ours [12, 13, 15, 16] aims to find counterfactual explanations that help individuals understand what would have to change for a predictive model to make a positive prediction about them. However, none of these works distinguish between decisions and predictions and, consequently, cannot be readily used to provide explanations to the decisions taken by a decision maker, which are ultimately what individuals who are subject to (semi)-automated decision making typically care about.

In our work, we build upon a recent line of work that explicitly distinguishes between predictions and decisions [17–22] and then pursue the development of methods to find counterfactual explanations for the decisions taken by a decision maker who is assisted by a data-driven predictive model. These counterfactual explanations help individuals understand what would have to change in order to receive a beneficial decision, rather than a positive prediction. Moreover, once we focus on explaining decisions, we cannot overlook the possibility that individuals may use these explanations to invest effort strategically in order to maximize their chances of receiving a beneficial decision. However, this is also an opportunity for us to find counterfactual explanations that help individuals to self-improve and eventually increase the utility of a decision policy, as noted by several studies in economics [23–25] and, more recently, in the computer science literature [21, 26, 27]. For example, if a bank explains to a customer that, if she reduces her credit card debt by 20%, she will receive the loan she is applying for, she may feel compelled to reduce her overall credit card debt by the proposed percentage to pay less interest, improving her financial situation, and this will eventually increase the profit the bank makes when she is able to successfully return the loan. This is in contrast with previous work on interpretable machine learning, which have ignored the influence that (counterfactual) explanations (of predictions by a predictive model) may have on the accuracy of predictive models and the utility of the decision policies[1].

**Our contributions.** We cast the above problem as a Stackelberg game in which the decision maker moves first and shares her counterfactual explanations before individuals best-respond to these explanations and invest effort to receive a beneficial decision. In this context, we assume that the decision maker takes decisions based on low dimensional feature vectors since, in many realistic scenarios, the data is summarized by just a small number of summary statistics (*e.g.*, FICO scores) [28, 29]. Under this problem formulation, we first show that, given a pre-defined policy, the problem of finding the optimal set of counterfactual explanations is NP-hard by using a novel reduction of the Set Cover problem [30]. Then, we show that the corresponding objective function is monotone and submodular and, as a direct consequence, it readily follows that a standard greedy algorithm offers approximation guarantees. In addition, we show that, given a pre-defined set of counterfactual explanations, the optimal policy is deterministic and can be computed in polynomial time. Moreover, building on this result, we can reduce the problem of jointly finding both the optimal policy and set of counterfactual explanations to maximizing a non-monotone submodular function. As a consequence, we can use a recent randomized algorithm to solve the problem, which also offers approximation guarantees. Further, we demonstrate that, by incorporating a matroid constraint into the problem formulation, we can increase the diversity of the optimal set of counterfactual explanations and incentivize individuals across the whole spectrum of the population to self improve. Experiments using real lending and credit card data illustrate our theoretical findings and show that the counterfactual explanations and decision policies found by the above algorithms achieve higher utility than several competitive baselines[2].

## 2 Problem Formulation

Given an individual with a feature vector $\boldsymbol{x} \in \{1, ..., n\}^d$ and a (*ground-truth*) label $y \in \{0, 1\}$, we assume a decision $d(\boldsymbol{x}) \in \{0, 1\}$ controls whether the corresponding label is *realized*[3]. This setting fits a variety of real-world scenarios, where continuous features are often discretized into (percentile) ranges. For example, in university admissions, the decision specifies whether a student is admitted ($d(\boldsymbol{x}) = 1$) or rejected ($d(\boldsymbol{x}) = 0$); the label indicates whether the student completes the program ($y = 1$) or drops out ($y = 0$) upon acceptance; and the feature vector ($\boldsymbol{x}$) may include her GRE scores, undergraduate GPA percentile, or research experience. Throughout the paper, we will denote the set of feature values as $\mathcal{X} = \{\boldsymbol{x}_1, \boldsymbol{x}_2, \ldots, \boldsymbol{x}_m\}$, where $m = n^d$ denotes the number of feature values, and assume that the number of features $d$ is small, as discussed previously.

Each decision is sampled from a decision policy $d(\boldsymbol{x}) \sim \pi(d \,|\, \boldsymbol{x})$, where, for brevity, we will write $\pi(\boldsymbol{x}) = \pi(d = 1 \,|\, \boldsymbol{x})$. For each individual, the label $y$ is sampled from a conditional probability distribution $y \sim P(y \,|\, \boldsymbol{x})$ and, without loss of generality, we index the feature values in decreasing order with respect to their corresponding outcome, *i.e.*, $i < j \Rightarrow P(y = 1 \,|\, \boldsymbol{x}_i) \geq P(y = 1 \,|\, \boldsymbol{x}_j)$. Moreover, we adopt a Stackelberg game-theoretic formulation in which each individual with initial feature value $\boldsymbol{x}_i$ receives a (counterfactual) explanation from the decision maker by means of a feature value $\mathcal{E}(\boldsymbol{x}_i) \in \mathcal{A} \subseteq \mathcal{P}_\pi := \{\boldsymbol{x} \in \mathcal{X} : \pi(\boldsymbol{x}) = 1\}$ before she (best-)responds[4]. This formulation fits a variety of real-world applications. For example, insurance companies often provide online car insurance simulators that, on the basis of a customer's initial feature value $\boldsymbol{x}_i$, let the customer know whether they are eligible for a particular deal. In case the customer does not qualify, the simulator could provide a counterfactual example $\mathcal{E}(\boldsymbol{x}_i)$ under which the individual is guaranteed to be eligible. In the remainder, we will refer to $\mathcal{A}$ as the set of counterfactual explanations and, for each individual with initial feature value $\boldsymbol{x}_i$, we will assume she does not know anything about the other counterfactual explanations $\mathcal{A} \backslash \mathcal{E}(\boldsymbol{x}_i)$ other individuals may receive nor the decision policy $\pi(\boldsymbol{x})$.

Now, let $c(\boldsymbol{x}, \mathcal{E}(\boldsymbol{x}_i))$ be the cost[5] an individual pays for changing from $\boldsymbol{x}_i$ to $\mathcal{E}(\boldsymbol{x}_i)$ and $b(\pi, \boldsymbol{x}) = \mathbb{E}_{d \sim \pi(d \,|\, x)}[d(\boldsymbol{x})]$ be the (immediate) benefit she obtains from a policy $\pi$, which is just the probability that the individual receives a positive decision. Then, following Tabibian et al. [21], each individual's best response is to change from her initial feature value $\boldsymbol{x}_i$ to $\mathcal{E}(\boldsymbol{x}_i)$ iff the gained benefit she would obtain outweighs the cost she would pay for changing features, *i.e.*,

$$\mathcal{E}(\boldsymbol{x}_i) \in \{\boldsymbol{x}_j \in \mathcal{X} \,:\, b(\pi, \boldsymbol{x}_j) - c(\boldsymbol{x}_i, \boldsymbol{x}_j) \geq b(\pi, \boldsymbol{x}_i)\} := \mathcal{R}(\boldsymbol{x}_i),$$

and it is to keep her initial feature value $\boldsymbol{x}_i$ otherwise. Here, we will refer to $\mathcal{R}(\boldsymbol{x}_i)$ as the *region of adaptation*. Then, at a population level, the above best response results into a transportation of mass between the original feature distribution $P(\boldsymbol{x})$ and a new feature distribution $P(\boldsymbol{x} \,|\, \pi, \mathcal{A})$ induced by the policy $\pi$ and the counterfactual explanations $\mathcal{A}$. More specifically, we can readily derive an analytical expression for the induced feature distribution in terms of the original feature distribution, *i.e.*, for all $\boldsymbol{x}_j \in \mathcal{X}$,

$$P(\boldsymbol{x}_j \,|\, \pi, \mathcal{A}) = P(\boldsymbol{x}_j)\mathbb{I}(\mathcal{R}(\boldsymbol{x}_j) \cap \mathcal{A} = \emptyset) + \sum_{i \in [m]} P(\boldsymbol{x}_i)\mathbb{I}(\mathcal{E}(\boldsymbol{x}_i) = \boldsymbol{x}_j \wedge \boldsymbol{x}_j \in \mathcal{R}(\boldsymbol{x}_i)),$$

Similarly as in previous work [17, 18, 21, 22], we will assume that the decision maker is rational, has access to (an estimation of) the original feature distribution $P(\boldsymbol{x})$, and aims to maximize the (immediate) utility $u(\pi, \gamma)$, which is the expected overall profit she obtains, *i.e.*,

$$\begin{aligned} u(\pi, \mathcal{A}) &= \mathbb{E}_{\boldsymbol{x} \sim P(\boldsymbol{x} \,|\, \pi, \mathcal{A}), y \sim P(y \,|\, \boldsymbol{x}), d \sim \pi(\boldsymbol{x})} \left[ yd(\boldsymbol{x}) - \gamma d(\boldsymbol{x}) \right] \\ &= \mathbb{E}_{\boldsymbol{x} \sim P(\boldsymbol{x} \,|\, \pi, \mathcal{A})} \left[ \pi(\boldsymbol{x})(P(y = 1 \,|\, \boldsymbol{x}) - \gamma) \right], \end{aligned} \qquad (1)$$

where $\gamma \in (0, 1)$ is a given constant reflecting economic considerations of the decision maker. For example, in university admissions, the term $\pi(\boldsymbol{x})P(y = 1 \,|\, \boldsymbol{x})$ is proportional to the expected number of students who are admitted and complete the program, the term $\pi(\boldsymbol{x})\gamma$ is proportional to the number of students who are admitted, and $\gamma$ measures the cost of education in units of graduated students. As a direct consequence, given a feature value $\boldsymbol{x}_i$ and a set of counterfactual explanations $\mathcal{A}$, we can conclude that, if $\mathcal{R}(\boldsymbol{x}_i) \cap \mathcal{A} \neq \emptyset$, the decision maker will decide to provide the counterfactual explanation $\mathcal{E}(\boldsymbol{x}_i)$ that provides the largest utility gain under the assumption that individuals best respond, *i.e.*,

$$\mathcal{E}(\boldsymbol{x}_i) = \underset{\boldsymbol{x} \in \mathcal{A} \cap \mathcal{R}(\boldsymbol{x}_i)}{\operatorname{argmax}} P(y \,|\, \boldsymbol{x}) \text{ for all } \boldsymbol{x}_i \in \mathcal{X} \setminus \mathcal{P}_\pi \text{ such that } \mathcal{R}(\boldsymbol{x}_i) \cap \mathcal{A} \neq \emptyset, \qquad (2)$$

and, if $\mathcal{R}(\boldsymbol{x}_i) \cap \mathcal{A} = \emptyset$, we arbitrarily assume that $\mathcal{E}(\boldsymbol{x}_i) = \operatorname{argmin}_{\boldsymbol{x} \in \mathcal{A}} c(\boldsymbol{x}_i, \boldsymbol{x})$[6].

Given the above preliminaries, our goal is to help the decision maker to first find the optimal set of counterfactual explanations $\mathcal{A}$ for a pre-defined policy in Section 3 and then both the optimal policy $\pi$ and set of counterfactual explanations $\mathcal{A}$ in Section 4.

**Remarks.** Given an individual with initial feature value $\boldsymbol{x}$, one may think that, by providing the counterfactual explanation $\mathcal{E}(\boldsymbol{x}) \in \mathcal{A} \cap \mathcal{R}(\boldsymbol{x})$ that gives the largest utility gain, the decision maker

is not acting in the individual's best interest but rather selfishly. This is because there may exist another counterfactual explanation $\mathcal{E}_m(\boldsymbol{x}) \in \mathcal{A} \cap \mathcal{R}(\boldsymbol{x})$ with lower cost for the individual, *i.e.*, $c(\boldsymbol{x}, \mathcal{E}_m(\boldsymbol{x})) \leq c(\boldsymbol{x}, \mathcal{E}(\boldsymbol{x}))$. In our work, we argue that the provided counterfactual explanations help the individual to achieve a greater self-improvement and this is likely to result in a superior long-term well-being, as illustrated in Figure 7(c) in Appendix E. For example, consider a bank issuing credit cards who wants to maintain credit for trustworthy customers and incentivize the more risky ones to improve their financial status. In this case, $\mathcal{E}(\boldsymbol{x})$ is the explanation that maximally improves the financial status of the individual, making the repayment more likely, but requires her to pay a larger (immediate) cost. In contrast, $\mathcal{E}_m(\boldsymbol{x})$ is an alternate explanation that requires the individual to pay a smaller (immediate) cost but, in comparison with $\mathcal{E}(\boldsymbol{x})$, would result in a higher risk of default. In this context, note that the individual would be "willing" to pay the cost of following either $\mathcal{E}(\boldsymbol{x})$ or $\mathcal{E}_m(\boldsymbol{x})$ since both explanations lie within the region of adaptation $\mathcal{R}(\boldsymbol{x})$. We refer the interested reader to Appendix F.2 for an anecdotal real-world example of $\mathcal{E}(\boldsymbol{x})$ and $\mathcal{E}_m(\boldsymbol{x})$.

As argued very recently [21, 26, 31], due to Goodhart's law, the conditional probability $P(y \mid \boldsymbol{x})$ may change after individuals (best)-respond if the true causal effect between the observed features $\boldsymbol{x}$ and the outcome variable $y$ is partially described by unobserved features. Moreover, Miller et al. [31] have argued that, to distinguish between gaming and improvement, it is necessary to have access to the full underlying causal graph between the features and the outcome variable. In this work, for simplicity, we assume that $P(y \mid \boldsymbol{x})$ does not change, however, it would be very interesting to lift this assumption in future work.

## 3 Finding the optimal counterfactual explanations for a policy

In this section, our goal is to find the optimal set of counterfactual explanations $\mathcal{A}^*$ for a pre-defined policy $\pi$, *i.e.*,

$$\mathcal{A}^* = \underset{\mathcal{A} \subseteq \mathcal{P}_\pi \,:\, |\mathcal{A}| \leq k}{\operatorname{argmax}} u(\pi, \mathcal{A}), \tag{3}$$

where the cardinality constraint on the set of counterfactual explanations balances the decision maker's obligation to be transparent with trade secrets [32]. More specifically, note that, without this constraint, an adversary could reverse-engineer the entire decision policy $\pi(\boldsymbol{x})$ by impersonating individuals with different feature values $\boldsymbol{x}$ [33].

As it will become clearer in the experimental evaluation in Section 6, our results may persuade decision makers to be transparent about their decision policies, something they are typically reluctant to be despite the increasing legal requirements, since we show that transparency increases the utility of the policies. Moreover, throughout this section, we will assume that the decision maker who picks the pre-defined policy is rational[7] and the policy is outcome monotonic[8][9] [21]. Outcome monotonicity just implies that, the higher an individual's outcome $P(y = 1 \mid \boldsymbol{x})$, the higher their chances of receiving a positive decision $\pi(\boldsymbol{x})$.

Unfortunately, using a novel reduction of the Set Cover problem [30], the following theorem reveals that we cannot expect to find the optimal set of counterfactual explanations in polynomial time (proven in Appendix B.1):

**Theorem 1** *The problem of finding the optimal set of counterfactual explanations that maximizes utility under a cardinality constraint is NP-Hard.*

Even though Theorem 1 is a negative result, we will now show that the objective function in Eq. 3 satisfies a set of desirable properties, *i.e.*, non-negativity, monotonicity and submodularity[10], which allow a standard greedy algorithm to enjoy approximation guarantees at solving the problem. To this aim, with a slight abuse of notation, we first express the objective function as a set function $f(\mathcal{A}) = u(\pi, \mathcal{A})$, which takes values over the ground set of counterfactual explanations, $\mathcal{P}_\pi$. Then, we have the following proposition (proven in Appendix B.2):

**Proposition 2** *The function $f$ is non-negative, submodular and monotone.*

The above result directly implies that the standard greedy algorithm [34] for maximizing a non-negative, submodular and monotone function will find a solution $\mathcal{A}$ to the problem such that $f(\mathcal{A}) \geq (1 - 1/e)f(\mathcal{A}^*)$, where $\mathcal{A}^*$ is the optimal set of counterfactual explanations. The algorithm starts from a solution set $\mathcal{A} = \emptyset$ and it iteratively adds to $\mathcal{A}$ the counterfactual explanation $\boldsymbol{x} \in \mathcal{P}_\pi \setminus \mathcal{A}$ that provides the maximum marginal difference $f(\mathcal{A} \cup \{\boldsymbol{x}\}) - f(\mathcal{A})$. Algorithm 1 in Appendix C provides a pseudocode implementation of the algorithm.

Finally, since the greedy algorithm computes the marginal difference of $f$ for at most $m$ elements per iteration and, following from the proof of Proposition 2, the marginal difference $f(\mathcal{A} \cup \{\boldsymbol{x}\}) - f(\mathcal{A})$ can be computed in $\mathcal{O}(m)$, then it immediately follows that, in our problem, the greedy algorithm has an overall complexity of $\mathcal{O}(km^2)$.

## 4 Finding the optimal policy and counterfactual explanations

In this section, our goal is to jointly find the optimal decision policy and set of counterfactual explanations $\mathcal{A}^*$, *i.e.*,

$$\pi^*, \mathcal{A}^* = \underset{(\pi, \mathcal{A}): \mathcal{A} \subseteq \mathcal{P}_\pi \wedge |\mathcal{A}| \leq k}{\operatorname{argmax}} u(\pi, \mathcal{A}) \tag{4}$$

where, similarly as in the previous section, $k$ is the maximum number of counterfactual explanations the decision maker is willing to provide to the population to balance the right to explanation with trade secrets. By jointly optimizing both the decision policy and the counterfactual explanations, we may obtain an additional gain in terms of utility in comparison with just optimizing for the set of counterfactual explanations given the optimal decision policy in a non-strategic setting, as shown in Figure 6 in Appendix D. Moreover, as we will show in the experimental evaluation in Section 6, this additional gain will be significant.

Similarly as in Section 3, we cannot expect to find the optimal policy and set of counterfactual explanations in polynomial time. More specifically, we have the following negative result, which easily follows from Proposition 4 and slightly extending the proof of Theorem 1:

**Theorem 3** *The problem of jointly finding both the optimal policy and the set of counterfactual explanations that maximize utility under a cardinality constraint is NP-hard.*

However, while the problem of finding both the policy and the set of counterfactual explanations appears significantly more challenging than the problem of finding just the set of counterfactual explanations given a pre-defined policy (refer to Eq. 3), the following proposition shows that the problem is not inherently *harder*. More specifically, for each possible set of counterfactual explanations, it shows that the policy that maximizes the utility can be easily computed (proven in Appendix B.3):

**Proposition 4** *Given a set of counterfactual explanations $\mathcal{A} \subseteq \mathcal{Y} := \{\boldsymbol{x} \in \mathcal{X} : P(y = 1 \,|\, \boldsymbol{x}) \geq \gamma\}$[11], the policy $\pi^*_{\mathcal{A}} = \operatorname{argmax}_{\pi: \mathcal{A} \subseteq \mathcal{P}_\pi} u(\pi, \mathcal{A})$ that maximizes the utility is deterministic and can be found in polynomial time,* i.e.,

$$\pi^*_{\mathcal{A}}(\boldsymbol{x}) = \begin{cases} 1 & \text{if } \boldsymbol{x} \in \mathcal{A} \vee \{\boldsymbol{x}' \in \mathcal{A} : P(y = 1 \,|\, \boldsymbol{x}') > P(y = 1 \,|\, \boldsymbol{x}) \wedge c(\boldsymbol{x}, \boldsymbol{x}') \leq 1\} = \emptyset \wedge \boldsymbol{x} \in \mathcal{Y} \\ 0 & \text{otherwise.} \end{cases} \tag{5}$$

The above result implies that, to set all the values of the optimal decision policy, we only need to perform $\mathcal{O}(km)$ comparisons. Moreover, it reveals that, in contrast with the non strategic setting, the optimal policy given a set of counterfactual explanations is not a deterministic threshold rule with a single threshold [17, 22], *i.e.*,

$$\pi(\boldsymbol{x}) = \begin{cases} 1 & \text{if } P(y = 1 \,|\, \boldsymbol{x}) \geq \gamma \\ 0 & \text{otherwise,} \end{cases} \tag{6}$$

but rather a more conservative deterministic decision policy that does not depend only on the outcome $P(y = 1 \,|\, \boldsymbol{x})$ and $\gamma$ but also on the cost individuals pay to change features. Moreover, we can build up on the above result to prove that the problem of finding the optimal decision policy and set of counterfactual explanations can be reduced to maximizing a non-monotone submodular function. To this aim, let $\pi^*_{\mathcal{A}}$ be the optimal policy induced by a given set of counterfactual explanations $\mathcal{A}$, as in

Proposition 4, and define the set function $h(\mathcal{A}) = u(\pi_{\mathcal{A}}^*, \mathcal{A})$ over the ground set $\mathcal{Y}$. Then, we have the following proposition (proven in Appendix B.4):

**Proposition 5** *The function $h$ is non-negative, submodular and non-monotone.*

Fortunately, there exist efficient algorithms with global approximation guarantees for maximizing a non-monotone submodular function under cardinality constraints. In our work, we use the randomized polynomial time algorithm by Buchbinder et al. [35], which can find a solution $\mathcal{A}$ such that $h(\mathcal{A}) \geq (1/e)h(\mathcal{A}^*)$, where $\mathcal{A}^*$ and $\pi_{\mathcal{A}^*}^*$ are the optimal set of counterfactual explanations and decision policy, respectively. The algorithm is just a randomized variation of the standard greedy algorithm. It starts from a solution set $\mathcal{A} = \emptyset$ and it iteratively adds one counterfactual explanation $\boldsymbol{x} \in \mathcal{Y} \backslash \mathcal{A}$. However, instead of greedily choosing the element $\boldsymbol{x}$ that provides the maximum marginal difference $h(\mathcal{A} \cup \{\boldsymbol{x}\}) - h(\mathcal{A})$, it sorts all the candidate elements with respect to their marginal difference and picks one at random among the top $k$. Algorithm 2 in Appendix C provides a pseudocode implementation of the algorithm.

Finally, since the above randomized algorithm has a complexity of $\mathcal{O}(km)$ and, following from the proof of Proposition 5, the marginal difference of $h$ can be computed in $\mathcal{O}(m)$, it readily follows that, in our problem, the algorithm has a complexity of $\mathcal{O}(km^2)$.

# 5 Increasing the diversity of the counterfactual explanations

In many cases, decision makers may like to ensure that individuals across the whole spectrum of the population are incentivized to self-improve. For example, in a loan scenario, the bank may use age group as a feature to estimate the probability that a customer repays the loan, however, it may like to deploy a decision policy that incentivizes individuals across all age groups in order to improve the financial situation of all. To this aim, the decision maker can increase the diversity of the optimal set of counterfactual explanations by incorporating a matroid constraint into the problem formulation, rather than a cardinality constraint.

Formally, consider disjoint sets $\mathcal{X}_1, \mathcal{X}_2, \ldots, \mathcal{X}_l$ such that $\bigcup_i \mathcal{X}_i = \mathcal{X}$ and integers $d_1, d_2, \ldots, d_l$ such that $k = \sum_i d_i$. Then, a partition matroid is the collection of sets $\{S \subseteq 2^{\mathcal{X}} : |S \cap \mathcal{X}_i| \leq d_i \ \forall i \in [l]\}$. In the loan example, the decision maker could search for a set of counterfactual explanations $\mathcal{A}$ within a partition matroid where each one of the $\mathcal{X}_i$'s corresponds to the feature values covered by each age group and $d_i = k/l \ \forall i \in [l]$. This way, the set of counterfactual explanations $\mathcal{A}$ would include explanations for every age group.

In this case, the decision maker could rely on a variety of polynomial time algorithms with global guarantees for submodular function maximization under matroid constraints, *e.g.*, the algorithm by Calinescu et al. [36].

# 6 Experiments

In this section, we evaluate Algorithms 1 and 2 using real loan and credit card data and show that the counterfactual explanations and decision policies found by our algorithms achieve higher utility than several competitive baselines. Appendix E contains additional experiments on synthetic data.

**Experimental setup.** We experiment with two publicly available datasets: (i) the *lending* dataset [37], which contains information about all accepted loan applications in LendingClub during the 2007-2018 period and (ii) the *credit* dataset [38], which contains information about a bank's credit card payoffs[12]. For each accepted loan applicant (or credit card holder), we use various demographic information and financial status indicators as features $\boldsymbol{x}$ and the current loan status (or credit payoff status) as label $y$. Appendix F.1 contains more details on the specific features we used in each dataset and also describes the procedure we followed to approximate $P(y \,|\, \boldsymbol{x})$.

To set the values of the cost function $c(\boldsymbol{x}_i, \boldsymbol{x}_j)$, we use the maximum percentile shift among actionable features[13], similarly as in Ustun et al. [13]. More specifically, let $\mathcal{L}$ be the set of actionable (numerical) features and $\bar{\mathcal{L}}$ be the set of non-actionable (discrete-valued) features[14]. Then, for each pair of feature values $\boldsymbol{x}_i, \boldsymbol{x}_j$ we define the cost function as:

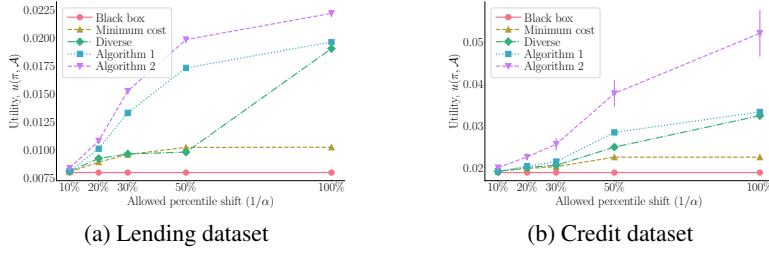

(a) Lending dataset            (b) Credit dataset

Figure 1: Utility achieved by five types of decision policies and counterfactual explanations against the value of the parameter $\alpha$, in the lending and credit datasets. In panel (a), the number of feature values is $m = 400$ and, in panel (b), it is $m = 3200$. In both panels, we set $k = 0.05m$ and we repeat each experiment 20 times.

$$c(\boldsymbol{x}_i, \boldsymbol{x}_j) = \begin{cases} \alpha \cdot \max_{l \in \mathcal{L}} |Q_l(x_{j,l}) - Q_l(x_{i,l})| & \text{if } x_{i,l} = x_{j,l} \; \forall l \in \bar{\mathcal{L}} \\ \infty & \text{otherwise,} \end{cases} \tag{7}$$

where $x_{j,l}$ is the value of the $l$-th feature for the feature value $\boldsymbol{x}_j$, $Q_l(\cdot)$ is the CDF of the numerical feature $l \in \mathcal{L}$ and $\alpha \geq 1$ is a scaling factor. As an exception, in the credit dataset, we always set the cost $c(\boldsymbol{x}_i, \boldsymbol{x}_j)$ between two feature values to $\infty$ if $Q_l(x_{j,l}) < Q_l(x_{i,l})$ for $l \in \{\text{Total Overdue Counts, Total Months Overdue}\}$ considering the fact that history of overdue payments cannot be erased. In this context, we would like to acknowledge that more sophisticated cost functions can be designed in terms of feasibility and difficulty of adaptation, taking into account domain knowledge and information about the deployed classifier, however, it goes beyond the scope of our work.

Finally, in our experiments, we compare the utility of the following decision policies and counterfactual explanations:

— *Black box:* decisions are taken by the optimal decision policy in the non-strategic setting, given by Eq. 6, and individuals do not receive any counterfactual explanations.

— *Minimum cost:* decisions are taken by the optimal decision policy in the non-strategic setting, given by Eq. 6, and individuals receive counterfactual explanations of minimum cost with respect to their initial feature values, similarly as in previous work [13, 15, 39]. More specifically, we cast the problem of finding the set of counterfactual explanations as the minimization of the weighted average cost individuals pay to change their feature values to the closest counterfactual explanation, *i.e.*,

$$\mathcal{A}_{mc} = \underset{\mathcal{A} \subseteq \mathcal{P}_\pi : |\mathcal{A}| \leq k}{\operatorname{argmin}} \sum_{\boldsymbol{x}_i \in \mathcal{X} \setminus \mathcal{P}_\pi} P(\boldsymbol{x}_i) \min_{\boldsymbol{x}_j \in \mathcal{A}} c(\boldsymbol{x}_i, \boldsymbol{x}_j),$$

and realize that this problem is a version of the k-median problem, which we can solve using a greedy heuristic [40].

— *Diverse:* decisions are taken by the optimal decision policy in the non-strategic setting, given by Eq. 6, and individuals receive a set of diverse counterfactual explanations of minimum cost with respect to their initial feature values, similarly as in previous work [16, 41], *i.e.*,

$$\mathcal{A}_d = \underset{\mathcal{A} \subseteq \mathcal{P}_\pi : |\mathcal{A}| \leq k}{\operatorname{argmax}} \sum_{\boldsymbol{x}_i \in \mathcal{X} \setminus \mathcal{P}_\pi} P(\boldsymbol{x}_i) \mathbb{I}(\mathcal{R}(\boldsymbol{x}_i) \cap \mathcal{A} \neq \emptyset),$$

To solve the above problem, we realize it can be reduced to the weighted version of the maximum coverage problem, which can be solved using a well-known greedy approximation algorithm [42].

— *Algorithm 1:* decisions are taken by the optimal decision policy in the non-strategic setting, given by Eq. 6, and individuals receive counterfactual explanations given by Eq. 2, where $\mathcal{A}$ is found using Algorithm 1.

— *Algorithm 2:* decisions are taken by the decision policy given by Eq. 5 and individuals receive counterfactual explanations given by Eq. 2, where $\mathcal{A}$ is found using Algorithm 2.

**Results.** We start by comparing the utility achieved by each of the decision policies and counterfactual explanations in both datasets, for several values of the parameter $\alpha$, which is proportional to the difficulty of changing features. Figure 1 summarizes the results, which show that Algorithm 1 and

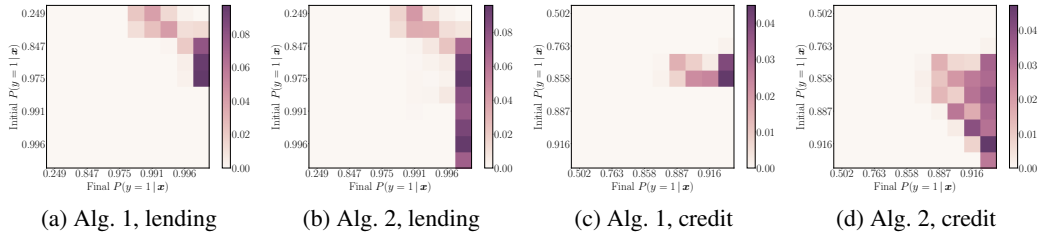

| (a) Alg. 1, lending | (b) Alg. 2, lending | (c) Alg. 1, credit | (d) Alg. 2, credit |

Figure 2: Transportation of mass induced by the policies and counterfactual explanations used in Algorithm 1 and 2 in both the lending and the credit dataset. For each individual in the population, whose best-response is to change her feature value, we record her outcome $P(y = 1 \mid \boldsymbol{x})$ before the best response (Initial $P(y = 1 \mid \boldsymbol{x})$) and after the best response (Final $P(y = 1 \mid \boldsymbol{x})$). In each panel, the color is proportional to the percentage of individuals who move from initial $P(y = 1 \mid \boldsymbol{x})$ to final $P(y = 1 \mid \boldsymbol{x})$ and we set $\alpha = 2$.

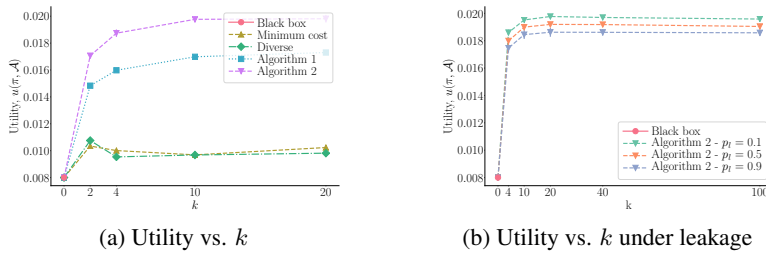

| (a) Utility vs. $k$ | (b) Utility vs. $k$ under leakage |

Figure 3: Number of counterfactual explanations and information leakage. Panel (a) shows the utility achieved by five types of decision policies and counterfactual explanations against the number of counterfactual explanations $k$. Panel (b) shows the utility achieved by Algorithm 2 against the number of counterfactual explanations $k$ for several values of the leakage probability $p_l$. In both panels, we use the lending dataset, the number of feature values is $m = 400$, we set $\alpha = 2$, we repeat each experiment involving randomization 20 times.

Algorithm 2 consistently outperform all baselines and, as the cost of adapting to feature values with higher outcome values decreases (smaller $\alpha$), the competitive advantage by jointly optimizing the decision policy and the counterfactual explanations (Algorithm 2) grows significantly. This competitive advantage is more apparent in the credit card dataset because it contains non actionable features (*e.g.*, credit overdue counts) and, under the optimal decision policy in the non-strategic setting, it is difficult to incentivize individuals who receive a negative decision to improve by just optimizing the set of counterfactual explanations they receive. For specific examples of counterfactual explanations provided by Algorithm 1 and the minimum cost baseline, refer to Appendix F.2.

To understand the differences in utility caused by the two proposed algorithms, we measure the transportation of mass induced by the policies and counterfactual explanations used in Algorithm 1 and 2 in both datasets, as follows. For each individual in the population whose best-response is to change her feature value, we record her outcome $P(y = 1 \mid \boldsymbol{x})$ before and after the best response. Then, we discretize the outcome values using percentiles. Figure 2 summarizes the results, which show several interesting insights. In the lending dataset, we observe that a large portion of individuals do improve their outcome even if we only optimize the counterfactual explanations (Panel (a)). In contrast, in the credit dataset, we observe that, if we only optimize the counterfactual explanations (Panel (c)), most individuals do not improve their outcome. That being said, if we jointly optimize the decision policy and counterfactual explanations (Panels (b) and (d)), we are able to incentivize a large portion of individuals to self improve in both datasets.

Next, we focus on the lending dataset and evaluate the sensitivity of our algorithms. First, we measure the influence that the number of counterfactual explanations has on the utility achieved by each of the decision policies and counterfactual explanations. As shown in Figure 3(a), our algorithms just need a small number of counterfactual explanations to provide significant gains in terms of utility with respect to all the baselines. Second, we challenge the assumption that individuals do not share the counterfactual explanations they receive with other individuals with different feature values. To this end, we assume that, given the set of counterfactual explanations $\mathcal{A}$ found by Algorithm 2, individuals with initial feature value $\boldsymbol{x}$ receive the counterfactual explanation $\mathcal{E}(\boldsymbol{x}) \in \mathcal{A}$ given by Eq. 2 and, with

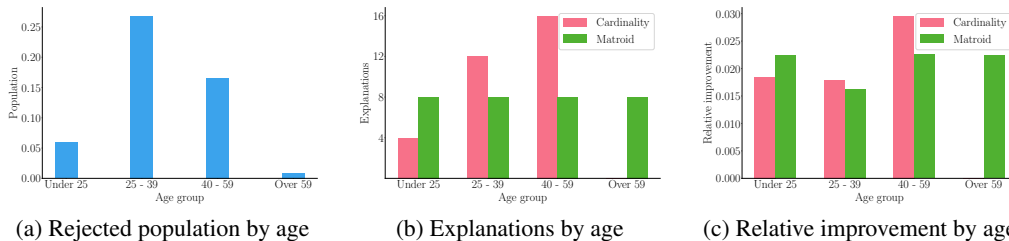

|     |     |     |
| --- | --- | --- |
| (a) Rejected population by age | (b) Explanations by age | (c) Relative improvement by age |

Figure 4: Increasing the diversity of the provided counterfactual explanations. Panel (a) shows the population per age group, rejected by the optimal threshold policy in the non strategic setting. Panel (b) shows a comparison of the age distribution of counterfactual explanations in $\mathcal{A}$ produced by the greedy algorithm under a cardinality and a matroid constraint. Panel (c) shows the relative improvement of each age group. In all panels, we use the credit dataset and we set $k = 32$ and $\alpha = 2$.

probability $p_l$, they also receive an additional explanation $\mathcal{E}'(\boldsymbol{x})$ picked at random from $\mathcal{A}$ and they follow the counterfactual explanation that benefits them the most. Figure 3(b) summarizes the results for several values of $p_l$ and number of counterfactual explanations, which show that the policies and explanations provided by Algorithm 2 present a significant utility advantage even when the leakage probability $p_l$ is large.

Finally, we focus on the credit dataset and consider a scenario in which a bank aims not only to continue providing credit to the customers that are more likely to repay but also provide explanations that incentivize individuals across all age groups to maintain their credit. To this end, we incorporate a partition matroid constraint that ensures the counterfactual explanations are diverse across age groups, as described in Section 5, and use a slightly modified version of Algorithm 1 to solve the constrained problem [34], which enjoys a $1/2$ approximation guarantee. Figure 4 summarizes the results, which show that: (i) optimizing under a cardinality constraint leads to an unbalanced set of explanations, favoring the more populated age groups (25 to 59) while completely ignoring the recourse potential of individuals older than 60; (ii) the relative group improvement, defined as $\sum_{\boldsymbol{x}_i \in \mathcal{X}_z \setminus \mathcal{P}_\pi} P(\boldsymbol{x}_i)[P(y \mid \boldsymbol{x}_j^i) - P(y \mid \boldsymbol{x}_i)] / \sum_{\boldsymbol{x}_i \in \mathcal{X}_z \setminus \mathcal{P}_\pi} P(\boldsymbol{x}_i)$, where $\mathcal{X}_z$ is the set of feature values corresponding to age group $z$ and $\boldsymbol{x}_j^i$ is the best response of individuals with initial feature value $\boldsymbol{x}_i \in \mathcal{X}_z$, is more balanced across age groups, showing that the matroid constraint can be used to generate counterfactual explanations that help the entire spectrum of the population to self-improve.

## 7 Conclusions

In this paper, we have designed several algorithms that allow us to find the decision policies and counterfactual explanations that maximize utility in a setting in which individuals who are subject to the decisions taken by the policies use the counterfactual explanations they receive to invest effort strategically. Moreover, we have experimented with synthetic and real lending and credit card data and shown that the counterfactual explanations and decision policies found by our algorithms achieve higher utility than several competitive baselines.

By uncovering a previously unexplored connection between strategic machine learning and interpretable machine learning, our work opens up many interesting directions for future work. For example, we have adopted a specific type of mechanism to provide counterfactual explanations (*i.e.*, one feature value per individual using a Stackelberg formulation). A natural next step would be to extend our analysis to other types of mechanisms fitting a variety of real-world applications. Moreover, we have assumed that the cost individuals pay to change features is given. However, our algorithms would be more effective if we develop a methodology to reliably estimate the cost function from real observational (or interventional) data. In our work, we have assumed that features take discrete values and individuals who are subject to the decisions do not share information between them. It would be interesting to lift these assumptions, extend our analysis to real-valued feature values, and develop decision policies and counterfactual explanations that are robust to information sharing between individuals (refer to Figure 3(c)). Finally, by assuming that $P(y \mid \boldsymbol{x})$ does not change after individuals best respond, we are implicitly assuming that there are not unobserved features that partially describe the true causal effect between the observed features $\boldsymbol{x}$ and the outcome variable $y$. However, in practice, this assumption is likely to be violated and $P(y \mid \boldsymbol{x})$ may change after individuals best respond, as recently noted by Miller et al. [31]. In this context, it would be very interesting to find counterfactual explanations that are robust to unmeasured confounding.

## 8   Broader Impact

In recent years, there has been a heated debate concerning the (lack of) transparency of machine learning models used to inform decisions that have significant consequences for individuals. In this context, decision makers are usually reluctant to share their decision policy with the individuals they decide upon because of trade secret concerns. In our work, we show that, if explanations for (semi-)automated decisions are chosen taking into account how individuals will best respond to them, the explanations can significantly increase the decision-maker's utility while also helping the individuals to self-improve, making transparency desirable for both sides.

## Acknowledgements

This project has received funding from the European Research Council (ERC) under the European Union's Horizon 2020 research and innovation programme (grant agreement No. 945719).

## Footnotes

[1]Refer to Appendix A for a discussion of further related work.

[2]An open-source implementation can be found at https://github.com/Networks-Learning/strategic-decisions.

[3]Without loss of generality, we assume each feature takes $n$ different values.

[4]In practice, individuals with initial feature values $\boldsymbol{x}_i$ such that $\pi(\boldsymbol{x}) = 1$ may not receive any explanation since they are guaranteed to receive a positive decision.

[5]In practice, the cost for each pair of feature values may be given by a parameterized function.

[6]Note that, if $\mathcal{A} \cap \mathcal{R}(\boldsymbol{x}_i) = \emptyset$, the individual's best response is to keep her initial feature value $\boldsymbol{x}_i$ and thus any choice of counterfactual explanation $\mathcal{E}(\boldsymbol{x}_i)$ leads to the same utility.

[7]Note that, if the decision maker is rational and her goal is to maximize the utility, as defined in Eq. 1, then, for all $\boldsymbol{x} \in \mathcal{X}$ such that $P(y = 1 \mid \boldsymbol{x}) < \gamma$, it holds that $\pi(\boldsymbol{x}) = 0$.

[8]A policy $\pi$ is called outcome monotonic if $P(y = 1 \mid \boldsymbol{x}_i) \geq P(y = 1 \mid \boldsymbol{x}_j) \Leftrightarrow \pi(\boldsymbol{x}_i) \geq \pi(\boldsymbol{x}_j) \; \forall \boldsymbol{x}_i, \boldsymbol{x}_j \in \mathcal{X}$.

[9]If the policy $\pi$ is deterministic, our results also hold for non outcome monotonic policies.

[10]A function $f : 2^{\mathcal{X}} \to \mathbb{R}$ is submodular if for every $\mathcal{A}, \mathcal{B} \subseteq \mathcal{X} : \mathcal{A} \subseteq \mathcal{B}$ and $x \in \mathcal{X} \setminus \mathcal{B}$ it holds that $f(\mathcal{A} \cup \{x\}) - f(\mathcal{A}) \geq f(\mathcal{B} \cup \{x\}) - f(\mathcal{B})$.

[11]Since the decision maker is rational, she will never provide an explanation that contributes negatively to her utility.

[12]We used a version of the credit dataset preprocessed by Ustun et al. [13]

[13]A feature is actionable if an individual can change its values in order to get a positive decision.

[14]In the credit dataset, $\bar{\mathcal{L}}$ contains Marital Status, Age Group and Education Level and $\bar{\mathcal{L}}$ contains the remaining features and, in the lending dataset, $\mathcal{L}$ contains all features.

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
