[Supplementary Material]

## A  Further related work

Our work builds upon previous work on interpretable machine learning, and strategic machine learning.

Most previous work on interpretable machine learning has focused on one of the two following types of explanations: feature-based explanations [8–10] or counterfactual explanations [12, 13, 15, 16]. Feature-based explanations help individuals understand the importance each feature has on a particular prediction, typically through local approximation, while counterfactual explanations help them understand what features would have to change for a predictive model to make a positive prediction about them. While there is not yet an agreement on what constitutes a *good* post-hoc explanation in the literature on interpretable machine learning, counterfactual explanations are gaining prominence because they place no constraints on the model complexity, do not require model disclosure, facilitate actionable recourse, and seem to automate compliance with the law [32]. Motivated by these desirable properties, our work focuses on counterfactual explanations and sheds light on the possibility of using explanations to increase the utility of a decision policy, uncovering a previously unexplored connection between interpretable machine learning and the nascent field of strategic machine learning.

Similarly as in our work, previous work on strategic machine learning also assumes that individuals may use knowledge, gained by transparency, to invest effort strategically in order to receive either a positive prediction [27, 31, 43–48] or a beneficial decision [21, 26]. However, none of this previous work focuses on finding (counterfactual) explanations and they assume full transparency—individuals who are subject to (semi)-automated decision making can observe the entire predictive model or the decision policy. As a result, their formulation is fundamentally different and their technical contributions are orthogonal to ours.

## B  Proofs

### B.1  Proof of Theorem 1

Consider an instance of the Set Cover problem with a set of elements $\mathcal{U} = \{u_1, \ldots, u_n\}$ and a collection $\mathcal{S} = \{\mathcal{S}_1, \ldots, \mathcal{S}_m\} \subseteq 2^{\mathcal{U}}$ such that $\bigcup_{i \in [m]} \mathcal{S}_i = \mathcal{U}$. In the decision version of the problem, given a constant $k$, we need to answer the question whether there are at most $k$ sets from the collection $\mathcal{S}$ such that their union is equal to $\mathcal{U}$ or not. With the following procedure, we show that any instance of that problem can be transformed to an instance of the problem of finding the optimal set of counterfactual explanations, defined in Eq. 3, in polynomial time.

Consider $n + m$ feature values corresponding to the $n$ elements of $\mathcal{U}$ and the $m$ sets of $\mathcal{S}$. Moreover, denote the first $n$ feature values as $\boldsymbol{x}_{u_1}, \ldots, \boldsymbol{x}_{u_n}$ and the remaining $m$ as $\boldsymbol{x}_{\mathcal{S}_1}, \ldots, \boldsymbol{x}_{\mathcal{S}_m}$. We set the decision maker's parameter $\gamma$ to some positive constant less than 1. Then, we set the outcome probabilities $P(y = 1 | \boldsymbol{x}_{u_i}) = \gamma \; \forall i \in [n]$ and $P(y = 1 | \boldsymbol{x}_{\mathcal{S}_i}) = 1 \; \forall i \in [m]$ and the policy values $\pi(\boldsymbol{x}_{u_i}) = 0 \; \forall i \in [n]$ and $\pi(\boldsymbol{x}_{\mathcal{S}_i}) = 1 \; \forall i \in [m]$. This way, the portion of utility the decision-maker obtains from the first $n$ feature values is zero, while the portion of utility she obtains from the remaining $m$ is proportional to $1 - \gamma$. Regarding the cost function, we set $c(\boldsymbol{x}_{u_i}, \boldsymbol{x}_{\mathcal{S}_j}) = 0 \; \forall (\boldsymbol{x}_{u_i}, \boldsymbol{x}_{\mathcal{S}_j}) : u_i \in S_j$, $c(\boldsymbol{x}_{u_i}, \boldsymbol{x}_{u_i}) = 0 \; \forall i \in [n]$, and all the remaining values of the cost function to 2. Finally, we set the initial feature value distribution to $P(\boldsymbol{x}_{u_i}) = \frac{1}{n} \; \forall i \in [n]$ and $P(\boldsymbol{x}_{\mathcal{S}_i}) = 0 \; \forall i \in [m]$. A toy example of this transformation is presented in Figure 5.

In this setting, it easy to observe that an individual with initial feature value $\boldsymbol{x}_{u_i}$ is always rejected at first and has the ability to move to a new feature value $\boldsymbol{x}_{\mathcal{S}_j}$ recommended to her iff $c(\boldsymbol{x}_{u_i}, \boldsymbol{x}_{\mathcal{S}_j}) \leq 1 \Leftrightarrow u_i \in \mathcal{S}_j$. Also, we can easily see that the transformation of instances can be done in $\mathcal{O}((m+n)^2)$ time.

Now, assume there exists an algorithm that optimally solves the problem of finding the optimal set of counterfactual explanations in polynomial time. Given the aforementioned instance and a maximum number of counterfactual explanations $k$, the utility $u(\pi, \mathcal{A})$ achieved by the set of counterfactual explanations $\mathcal{A}$ the algorithm returns can fall into one of the following two cases:

1. $u(\pi, \mathcal{A}) = 1 - \gamma$. This can happen only if all individuals, according to the induced distribution $P(\boldsymbol{x} \,|\, \pi, \mathcal{A})$, have moved to some of the feature values $\boldsymbol{x}_{\mathcal{S}_j}$, *i.e.*, for all $\boldsymbol{x}_{u_i}$ with $i \in [n]$, there exists $\boldsymbol{x}_{\mathcal{S}_j}$ with $j \in [m]$ such that $\boldsymbol{x}_{\mathcal{S}_j} \in \mathcal{A} \wedge c(\boldsymbol{x}_{u_i}, \boldsymbol{x}_{\mathcal{S}_j}) \leq 1$ with $|\mathcal{A}| \leq k$. As a consequence, if we define $\mathcal{S}' = \{\mathcal{S}_j : \boldsymbol{x}_{\mathcal{S}_j} \in \mathcal{A}\}$, it holds that for all $u_i$ with $i \in [n]$,

Figure 5: Consider that $\mathcal{U} = \{u_1, u_2\}$ and $\mathcal{S} = \{\mathcal{S}_1, \mathcal{S}_2\}$ with $\mathcal{S}_1 = \{u_1, u_2\}$, $\mathcal{S}_2 = \{u_2\}$. The red feature values have initial population $P(\boldsymbol{x}) = 1/2$, $\pi(\boldsymbol{x}) = 0$ and $P(y = 1 \mid \boldsymbol{x}) = \gamma$ while for the green feature values it is $P(\boldsymbol{x}) = 0$, $\pi(\boldsymbol{x}) = 1$ and $P(y = 1 \mid \boldsymbol{x}) = 1$. The edges represent the cost between feature values corresponding to sets and their respective elements while all the non-visible pairwise costs are equal to 2.

      there exists $\mathcal{S}_j$ with $j \in [m]$ such that $\mathcal{S}_j \in \mathcal{S}' \wedge u_i \in \mathcal{S}_j$ and therefore $\mathcal{S}'$ is a set cover with $|\mathcal{S}'| = |\mathcal{A}| \leq k$.

2. $u(\pi, \mathcal{A}) < 1 - \gamma$. This can happen only if every possible set of $k$ counterfactual explanations leaves the individuals of at least one feature value $\boldsymbol{x}_{u_i}$ with a best-response of not following the counterfactual explanation they were given, *i.e.*, for all $\mathcal{A} \subseteq \mathcal{P}_\pi$ such that $|\mathcal{A}| \leq k$, there exists $\boldsymbol{x}_{u_i}$ with $i \in [n]$ such that, for all $\boldsymbol{x}_{\mathcal{S}_j} \in \mathcal{A}$, it holds that $c(\boldsymbol{x}_{u_i}, \boldsymbol{x}_{\mathcal{S}_j}) > 1$. Equivalently, it holds that for all $\mathcal{S}' \subseteq \mathcal{S}$ such that $|\mathcal{S}'| \leq k$, there exists $u_i$ with $i \in [n]$ such that for all $\mathcal{S}_j \in \mathcal{S}'$, it holds that $u_i \notin \mathcal{S}_j$ and therefore there does not exist a set cover of size less or equal than $k$.

The above directly implies that we can have a decision about any instance of the Set Cover problem in polynomial time, which is a contradiction unless $P = NP$. This concludes the reduction and proves that the problem of finding the optimal set of counterfactual explanations for a given policy is NP-Hard.

## B.2 Proof of Proposition 2

It readily follows that the function $f$ is non-negative from the fact that, if the decision maker is rational, it holds that $\pi(\boldsymbol{x}) = 0$ for all $\boldsymbol{x} \in \mathcal{X}$ such that $P(y = 1 \mid \boldsymbol{x}) < \gamma$.

Now, consider two sets $\mathcal{A}, \mathcal{B} \subseteq \mathcal{P}_\pi : \mathcal{A} \subseteq \mathcal{B}$ and a feature value $\boldsymbol{x} \in \mathcal{P}_\pi \setminus \mathcal{B}$. Also, let $\mathcal{E}_\mathcal{S}(\boldsymbol{x}_i)$ be the counterfactual explanation given to the individuals with initial feature value $\boldsymbol{x}_i$ under a set of counterfactual explanations $\mathcal{S}$. It is easy to see that the marginal difference $f(\mathcal{S} \cup \{\boldsymbol{x}\}) - f(\mathcal{S})$ can only be affected by individuals with initial features $\boldsymbol{x}_i$ such that $\boldsymbol{x}_i \notin \mathcal{P}_\pi$, $\boldsymbol{x} \in \mathcal{R}(\boldsymbol{x}_i)$ and $\boldsymbol{x} = \mathcal{E}_{S \cup \{\boldsymbol{x}\}}(\boldsymbol{x}_i)$. Moreover, we can divide all of these individuals into two cases:

1. $\mathcal{R}(\boldsymbol{x}_i) \cap \mathcal{A} = \emptyset$: in this case, the addition of $\boldsymbol{x}$ to $\mathcal{A}$ causes a change in their best-response from $\boldsymbol{x}_i$ to $\boldsymbol{x}$ contributing to the marginal difference of $f$ by a factor $P(\boldsymbol{x}_i)[P(y = 1 \mid \boldsymbol{x}) - \gamma - \pi(\boldsymbol{x}_i)(P(y = 1 \mid \boldsymbol{x}_i) - \gamma)]$. However, considering the marginal difference of $f$ under the set of counterfactual explanations $\mathcal{B}$, three subcases are possible:

   (a) $\mathcal{E}_\mathcal{B}(\boldsymbol{x}_i) \in \mathcal{R}(\boldsymbol{x}_i) \wedge P(y = 1 \mid \mathcal{E}_\mathcal{B}(\boldsymbol{x}_i)) > P(y = 1 \mid \boldsymbol{x})$: the contribution to the marginal difference of $f$ is zero.

   (b) $\mathcal{E}_\mathcal{B}(\boldsymbol{x}_i) \in \mathcal{R}(\boldsymbol{x}_i) \wedge P(y = 1 \mid \mathcal{E}_\mathcal{B}(\boldsymbol{x}_i)) \leq P(y = 1 \mid \boldsymbol{x})$: the contribution to the marginal difference of $f$ is $P(\boldsymbol{x}_i)[P(y = 1 \mid \boldsymbol{x}) - P(y = 1 \mid \mathcal{E}_\mathcal{B}(\boldsymbol{x}_i))]$. Since $\pi$ is outcome monotonic, $\mathcal{E}_\mathcal{B}(\boldsymbol{x}_i) \in \mathcal{P}_\pi$ and $\boldsymbol{x}_i \notin \mathcal{P}_\pi$, it holds that

   $$P(y = 1 \mid \mathcal{E}_\mathcal{B}(\boldsymbol{x}_i)) \geq P(y = 1 \mid \boldsymbol{x}_i) \Rightarrow$$
   $$P(y = 1 \mid \mathcal{E}_\mathcal{B}(\boldsymbol{x}_i)) - \gamma \geq P(y = 1 \mid \boldsymbol{x}_i) - \gamma > \pi(\boldsymbol{x}_i)[P(y = 1 \mid \boldsymbol{x}_i) - \gamma].$$

   Therefore, it readily follows that

   $$P(\boldsymbol{x}_i)[P(y = 1 \mid \boldsymbol{x}) - P(y = 1 \mid \mathcal{E}_\mathcal{B}(\boldsymbol{x}_i))] <$$
   $$P(\boldsymbol{x}_i)[P(y = 1 \mid \boldsymbol{x}) - \gamma - \pi(\boldsymbol{x}_i)(P(y = 1 \mid \boldsymbol{x}_i) - \gamma)].$$

(c) $\mathcal{R}(\boldsymbol{x}_i) \cap \mathcal{B} = \emptyset$: the contribution to the marginal difference of $f$ is $P(\boldsymbol{x}_i)[P(y = 1 \,|\, \boldsymbol{x}) - \gamma - \pi(\boldsymbol{x}_i)(P(y = 1 \,|\, \boldsymbol{x}_i) - \gamma)]$.

2. $\mathcal{R}(\boldsymbol{x}_i) \cap \mathcal{A} \neq \emptyset \wedge P(y = 1 \,|\, \boldsymbol{x}) > P(y = 1 \,|\, \mathcal{E}_{\mathcal{A}}(\boldsymbol{x}_i))$: In this case, the addition of $\boldsymbol{x}$ to $\mathcal{A}$ causes a change in their best-response from $\mathcal{E}_{\mathcal{A}}(\boldsymbol{x}_i)$ to $\boldsymbol{x}$ contributing to the marginal difference of $f$ by a factor $P(\boldsymbol{x}_i)[P(y = 1 \,|\, \boldsymbol{x}) - P(y = 1 \,|\, \mathcal{E}_{\mathcal{A}}(\boldsymbol{x}_i))]$. Considering the marginal difference of $f$ under the set of counterfactual explanations $\mathcal{B}$, two subcases are possible:

    (a) $\mathcal{E}_{\mathcal{B}}(\boldsymbol{x}_i) \in \mathcal{R}(\boldsymbol{x}_i) \wedge P(y = 1 \,|\, \mathcal{E}_{\mathcal{B}}(\boldsymbol{x}_i)) > P(y = 1 \,|\, \boldsymbol{x})$: the contribution to the marginal difference of $f$ is zero.

    (b) $\mathcal{E}_{\mathcal{B}}(\boldsymbol{x}_i) \in \mathcal{R}(\boldsymbol{x}_i) \wedge P(y = 1 \,|\, \mathcal{E}_{\mathcal{B}}(\boldsymbol{x}_i)) \leq P(y = 1 \,|\, \boldsymbol{x})$. Then, the contribution of those individuals to the marginal difference of $f$ is $P(\boldsymbol{x}_i)[P(y = 1 \,|\, \boldsymbol{x}) - P(y = 1 \,|\, \mathcal{E}_{\mathcal{B}}(\boldsymbol{x}_i))]$. Since $\mathcal{A} \subseteq \mathcal{B}$ and $\mathcal{R}(\boldsymbol{x}_i) \cap \mathcal{A} \neq \emptyset$, it readily follows that

    $$P(y = 1 \,|\, \mathcal{E}_{\mathcal{B}}(\boldsymbol{x}_i)) \geq P(y = 1 \,|\, \mathcal{E}_{\mathcal{A}}(\boldsymbol{x}_i)) \Rightarrow$$
    $$P(\boldsymbol{x}_i)[P(y = 1 \,|\, \boldsymbol{x}) - P(y = 1 \,|\, \mathcal{E}_{\mathcal{A}}(\boldsymbol{x}_i))] \geq$$
    $$P(\boldsymbol{x}_i)[P(y = 1 \,|\, \boldsymbol{x}) - P(y = 1 \,|\, \mathcal{E}_{\mathcal{B}}(\boldsymbol{x}_i))].$$

Finally, because $\mathcal{A} \subseteq \mathcal{B}$, we can conclude that $f(\mathcal{B} \cup \{\boldsymbol{x}\}) - f(\mathcal{B}) \neq 0 \Rightarrow f(\mathcal{A} \cup \{\boldsymbol{x}\}) - f(\mathcal{A}) \neq 0$ and therefore the aforementioned cases are sufficient. Combining all cases, we can see that the contribution of each individual to the marginal difference of $f$ is always greater or equal under the set of counterfactual explanations $\mathcal{A}$ than under the set of counterfactual explanations $\mathcal{B}$. As a direct consequence, it follows that $f$ is submodular. Additionally, we can easily see that this contribution is always greater or equal than zero, leading to the conclusion that $f$ is also monotone.

## B.3 Proof of Proposition 4

By definition, since $\mathcal{A} \subseteq \mathcal{P}_{\pi_{\mathcal{A}}^*}$, it readily follows that $\pi_{\mathcal{A}}^*(\boldsymbol{x}) = 1$ for all $\boldsymbol{x} \in \mathcal{A}$. To find the remaining values of the decision policy, we first observe that, for each $\boldsymbol{x} \notin \mathcal{A}$, the value of the decision policy $\pi_{\mathcal{A}}^*(\boldsymbol{x})$ does not affect the best-responses of the individuals with initial feature values $\boldsymbol{x}' \neq \boldsymbol{x}$. As a result, we can just set $\pi_{\mathcal{A}}^*(\boldsymbol{x})$ for all $\boldsymbol{x} \notin \mathcal{A}$ independently for each feature value $\boldsymbol{x}$ such that the best-response of the respective individuals is the one that contributes maximally to the overall utility.

First, it is easy to see that, for all $\boldsymbol{x} \notin \mathcal{A}$ such that $P(y = 1 \,|\, \boldsymbol{x}) < \gamma$, we should set $\pi_{\mathcal{A}}^*(\boldsymbol{x}) = 0$. Next, consider the feature values $\boldsymbol{x} \notin \mathcal{A}$ such that $P(y = 1 \,|\, \boldsymbol{x}) \geq \gamma$. Here, we distinguish two cases. If there exists $\boldsymbol{x}' \in \mathcal{A}$ such that $c(\boldsymbol{x}, \boldsymbol{x}') \leq 1 \wedge P(y = 1 \,|\, \boldsymbol{x}') > P(y = 1 \,|\, \boldsymbol{x})$, then, if the individuals move to that $\boldsymbol{x}'$, the corresponding contribution to the utility will be higher. Moreover, the value of the decision policy that maximizes their region of adaption (and thus increases their chances of moving to $\boldsymbol{x}'$) is clearly $\pi_{\mathcal{A}}^*(\boldsymbol{x}) = 0$. If there does not exist $\boldsymbol{x}' \in \mathcal{A}$ such that $c(\boldsymbol{x}, \boldsymbol{x}') \leq 1 \wedge P(y = 1 \,|\, \boldsymbol{x}') > P(y = 1 \,|\, \boldsymbol{x})$, then, the contribution of the corresponding individuals to the utility will be higher if they keep their initial feature values. Moreover, the value of the decision policy that will maximize this contribution will be clearly $\pi_{\mathcal{A}}^*(\boldsymbol{x}) = 1$.

## B.4 Proof of Proposition 5

It readily follows that the function $h$ is non-negative from the fact that, if the decision maker is rational, $\pi(\boldsymbol{x}) = 0$ for all $\boldsymbol{x} \in \mathcal{X}$ such that $P(y = 1 \,|\, \boldsymbol{x}) < \gamma$.

Next, consider two sets $\mathcal{A}, \mathcal{B} \subseteq \mathcal{Y}$ such that $\mathcal{A} \subseteq \mathcal{B}$ and a feature value $\boldsymbol{x} \in \mathcal{Y} \setminus \mathcal{B}$. Also, let $\mathcal{E}_{\mathcal{S}}(\boldsymbol{x}_i)$ be the counterfactual explanation given to the individuals with initial feature value $\boldsymbol{x}_i$ under a set of counterfactual explanations $\mathcal{S}$. Then, it is clear that the marginal difference $h(\mathcal{S} \cup \{\boldsymbol{x}\}) - h(\mathcal{S})$ only depends on individuals with initial features $\boldsymbol{x}_i$ such that either $1 - c(\boldsymbol{x}_i, \boldsymbol{x}) \geq 0$ and $\boldsymbol{x} = \mathcal{E}_{\mathcal{S} \cup \{\boldsymbol{x}\}}(\boldsymbol{x}_i)$ or $\boldsymbol{x}_i = \boldsymbol{x}$. Moreover, if $1 - c(\boldsymbol{x}_i, \boldsymbol{x}) \geq 0$ and $\boldsymbol{x} = \mathcal{E}_{\mathcal{S} \cup \{\boldsymbol{x}\}}(\boldsymbol{x}_i)$, the contribution to the marginal difference is positive and, if $\boldsymbol{x}_i = \boldsymbol{x}$, the contribution to the marginal difference is negative.

Consider first the individuals with initial features $\boldsymbol{x}_i$ such that $1 - c(\boldsymbol{x}_i, \boldsymbol{x}) \geq 0$ and $\boldsymbol{x} = \mathcal{E}_{\mathcal{A} \cup \{\boldsymbol{x}\}}(\boldsymbol{x}_i)$. We can divide all of these individuals into three cases:

1. $\pi_{\mathcal{B}}(\boldsymbol{x}_i) = 0$: in this case, $\boldsymbol{x}_i \notin \mathcal{B}$ and the individuals change their best-response from $\mathcal{E}_{\mathcal{B}}(\boldsymbol{x}_i)$ to $\boldsymbol{x}$. Moreover, under the set of counterfactual explanations $\mathcal{A}$, their best-response is either $\boldsymbol{x}_i$ or $\mathcal{E}_{\mathcal{A}}(\boldsymbol{x}_i)$ and it changes to $\boldsymbol{x}$. Then, using a similar argument as in the proof

of proposition 2, we can conclude that the contribution of the individuals to the marginal difference is greater or equal under the set of counterfactual explanations $\mathcal{A}$ than under $\mathcal{B}$.

2. $\pi_{\mathcal{B}}(\boldsymbol{x}_i) = 1 \wedge \pi_{\mathcal{A}}(\boldsymbol{x}_i) = 0$: in this case, $\boldsymbol{x}_i \notin \mathcal{A}$ and $\boldsymbol{x}_i \in \mathcal{B}$. Therefore, under the set of counterfactual explanations $\mathcal{A}$, the individuals' best-response changes from $\mathcal{E}_{\mathcal{A}}(\boldsymbol{x}_i)$ to $\boldsymbol{x}$ and there is a positive contribution to the marginal difference while, under $\mathcal{B}$, the individuals' best response does not change and the contribution to the marginal difference is zero.

3. $\pi_{\mathcal{B}}(\boldsymbol{x}_i) = 1 \wedge \pi_{\mathcal{A}}(\boldsymbol{x}_i) = 1$: in this case, $\boldsymbol{x}_i \notin \mathcal{B}$. Therefore, the best-response changes from $\boldsymbol{x}_i$ to $\boldsymbol{x}$ under both sets of counterfactual explanations and there is an equal positive contribution to the marginal difference.

Now, consider the individuals with initial features $\boldsymbol{x}_i$ such that $\boldsymbol{x}_i = \boldsymbol{x}$. We can divide all of these individuals also into three cases:

1. $\pi_{\mathcal{A}}(\boldsymbol{x}) = \pi_{\mathcal{B}}(\boldsymbol{x}) = 0$: in this case, under both sets of counterfactual explanations, the counterfactual explanation $\boldsymbol{x}$ changes the value of the decision policy to $\pi_{\mathcal{A} \cup \{\boldsymbol{x}\}}(\boldsymbol{x}) = \pi_{\mathcal{B} \cup \{\boldsymbol{x}\}}(\boldsymbol{x}) = 1$. Moreover, the contribution to the marginal difference is less negative under the set of counterfactual explanations $\mathcal{A}$ than under $\mathcal{B}$ since $P(y = 1 \,|\, \mathcal{E}_{\mathcal{A}}(\boldsymbol{x})) \leq P(y = 1 \,|\, \mathcal{E}_{\mathcal{B}}(\boldsymbol{x}))$ and thus $P(\boldsymbol{x})[P(y = 1 \,|\, \boldsymbol{x}) - P(y = 1 \,|\, \mathcal{E}_{\mathcal{A}}(\boldsymbol{x}))] \geq P(\boldsymbol{x})[P(y = 1 \,|\, \boldsymbol{x}) - P(y = 1 \,|\, \mathcal{E}_{\mathcal{B}}(\boldsymbol{x}))]$.

2. $\pi_{\mathcal{A}}(\boldsymbol{x}) = 1 \wedge \pi_{\mathcal{B}}(\boldsymbol{x}) = 0$: in this case, under the set of counterfactual explanations $\mathcal{A}$, the individuals' best response does not change and thus the contribution to the marginal difference is zero and, under the set of counterfactual explanations $\mathcal{B}$, their best-response changes from $\mathcal{E}_{\mathcal{B}}(\boldsymbol{x})$ to $\boldsymbol{x}$ and thus there is a negative contribution to the marginal difference i.e., $P(\boldsymbol{x})[P(y = 1 \,|\, \boldsymbol{x}) - P(y = 1 \,|\, \mathcal{E}_{\mathcal{B}}(\boldsymbol{x}))] < 0$.

3. $\pi_{\mathcal{A}}(\boldsymbol{x}) = \pi_{\mathcal{B}}(\boldsymbol{x}) = 1$: in this case, under both sets of counterfactual explanations, the individuals' best response does not change and thus the contribution to the marginal difference is zero.

As a direct consequence of the above observations, it readily follows that $h(\mathcal{A} \cup \{\boldsymbol{x}\}) - h(\mathcal{A}) \geq h(\mathcal{B} \cup \{\boldsymbol{x}\}) - h(\mathcal{B})$ and therefore the function $h$ is submodular.

However, in contrast with Section 3, the function $h$ is non-monotone since it can happen that the negative marginal contribution exceeds the positive one. For example, consider the following instance of the problem, where $\boldsymbol{x} \in \{1, 2, 3\}$ with $\gamma = 0.1$:

$$P(\boldsymbol{x}) = 0.1\,\mathbb{I}(\boldsymbol{x} = 1) + 0.8\,\mathbb{I}(\boldsymbol{x} = 2) + 0.1\,\mathbb{I}(\boldsymbol{x} = 3),$$
$$P(y = 1 \,|\, \boldsymbol{x}) = 1.0\,\mathbb{I}(\boldsymbol{x} = 1) + 0.5\,\mathbb{I}(\boldsymbol{x} = 2) + 0.4\,\mathbb{I}(\boldsymbol{x} = 3),$$

and

$$c(\boldsymbol{x}_i, \boldsymbol{x}_j) = \begin{bmatrix} 0.0 & 0.2 & 0.3 \\ 0.3 & 0.0 & 0.7 \\ 0.4 & 0.5 & 0.0 \end{bmatrix}.$$

Assume there is a set of counterfactual explanations $\mathcal{A} = \{1\}$. Then, the optimal policy is given by $\pi_{\mathcal{A}}^*(1) = 1, \pi_{\mathcal{A}}^*(2) = 0, \pi_{\mathcal{A}}^*(3) = 0$ inducing a movement from feature values $2, 3$ to feature value $1$, giving a utility equal to $0.9$. Now, add $\boldsymbol{x} = 2$ to the set of counterfactual explanations i.e., $\mathcal{A} = \{1, 2\}$. Then, the optimal policy is given by $\pi_{\mathcal{A}}^*(1) = 1, \pi_{\mathcal{A}}^*(2) = 1, \pi_{\mathcal{A}}^*(3) = 0$ inducing a movement from feature value $3$ to feature value $1$, giving a lower utility, equal to $0.5$. Therefore, the function $h$ is non-monotone.

## C  Additional details on the standard greedy algorithm and the randomized algorithm by Buchbinder et al. [35]

To enjoy a $1/e$ approximation guarantee, Algorithm 2 requires that there are $2k < m$ candidate feature values whose marginal contribution to any set is zero. In our problem, this can be trivially satisfied by adding $2k$ feature values $\boldsymbol{x}$ to $\mathcal{X}$ such that $P(y = 1 \,|\, \boldsymbol{x}) = \gamma$, $P(\boldsymbol{x}) = 0$ and $c(\boldsymbol{x}, \boldsymbol{x}_j) = c(\boldsymbol{x}_j, \boldsymbol{x}) = 2 \;\forall \boldsymbol{x}_j \in \mathcal{X}$. If the algorithm adds some of those counterfactual explanations to the set $\mathcal{A}$, it is easy to see that we can ignore them without causing any difference in utility or best-responses.

**ALGORITHM 1:** Standard greedy algorithm [34]

**Input:** Ground set of counterfactual explanations $\mathcal{P}_\pi$, parameter $k$ and utility function $f$
**Output:** Set of counterfactual explanations $\mathcal{A}$
 1: $\mathcal{A} \leftarrow \varnothing$
 2: **while** $|\mathcal{A}| \leq k$ **do**
 3:    $\boldsymbol{x}^* \leftarrow \operatorname{argmax}_{\boldsymbol{x} \in \mathcal{P}_\pi \setminus \mathcal{A}} f(\mathcal{A} \cup \{\boldsymbol{x}\}) - f(\mathcal{A})$
 4:    $\mathcal{A} \leftarrow \mathcal{A} \cup \{\boldsymbol{x}^*\}$
 5: **end while**
 6: **return** $\mathcal{A}$

**ALGORITHM 2:** Randomized algorithm by Buchbinder et al. [35]

**Input:** Ground set of counterfactual explanations $\mathcal{Y}$, parameter $k$ and utility function $f$
**Output:** Set of counterfactual explanations $\mathcal{A}$
 1: $\mathcal{A} \leftarrow \varnothing$
 2: **while** $|\mathcal{A}| \leq k$ **do**
 3:    $\mathcal{B} \leftarrow \operatorname{GetTopK}(\mathcal{Y}, \mathcal{A}, f)$
 4:    $\boldsymbol{x}^* \sim \mathcal{B}$
 5:    $\mathcal{A} \leftarrow \mathcal{A} \cup \{\boldsymbol{x}^*\}$
 6: **end while**
 7: **return** $\mathcal{A}$

# D   Jointly optimizing the decision policy and the counterfactual explanations

Figure 6 shows that, by jointly optimizing both the decision policy and the counterfactual explanations, we may obtain an additional gain in terms of utility in comparison with just optimizing for the set of counterfactual explanations given the optimal decision policy in a non-strategic setting.

Non-strategic policy              Strategic policy

Figure 6: Jointly optimizing the decision policy and the counterfactual explanations can offer additional gains. The left panel shows the optimal (deterministic) decision policy $\pi$ under non-strategic behavior, as given by Eq. 6. Here, there does not exist a set of counterfactual explanations $\mathcal{A} \in \mathcal{P}_\pi$ that increases the utility of the policy. This happens because the area of adaption of $\boldsymbol{x}_3$ and $\boldsymbol{x}_4$ does not include any feature value that receives a positive decision. The right panel shows the decision policy and counterfactual explanations that are (jointly) optimal in terms of utility, as given by Eq. 4. Here, the individuals with feature values $\boldsymbol{x}_1$ and $\boldsymbol{x}_2$ receive $\mathcal{E}(\boldsymbol{x}_1)$ and $\mathcal{E}(\boldsymbol{x}_2)$, respectively, as counterfactual explanations. Since these explanations are within their areas of adaptation $\mathcal{R}(\boldsymbol{x}_1)$ and $\mathcal{R}(\boldsymbol{x}_2)$, they change their initial feature values in order to receive a positive decision.

# E   Experiments on Synthetic Data

**Experimental setup.** For simplicity, we consider feature values $\boldsymbol{x} \in \{0, \ldots, m-1\}$ and $P(\boldsymbol{x} = i) = p_i / \sum_j p_j$ where $p_i$ is sampled from a Gaussian distribution $N(\mu = 0.5, \sigma = 0.1)$ truncated from below at zero. We also sample $P(y = 1 \mid \boldsymbol{x}) \sim U[0, 1]$, $c(\boldsymbol{x}_i, \boldsymbol{x}_j) \sim U[0, 1]$ for 50% of all pairs and $c(\boldsymbol{x}_i, \boldsymbol{x}_j) = 2$ for the rest. Finally, we set $\gamma = 0.3$. In this section, we compare the utility achieved by our explanation methods with the same baselines we used on real data.

(a) Utility vs. # feature values     (b) Utility vs. # explanations     (c) Individual cost vs. # explanations

Figure 7: Results on synthetic data. Panels (a) and (b) show the utility achieved by six types of decision policies and counterfactual explanations against the total number of feature values $m$ and the number of counterfactual explanations $k$, respectively. Panel (c) shows the average cost individuals had to pay to change from their initial features to the feature value of the counterfactual explanation they receive under the same five types of decision policies and counterfactual explanations. In Panel (a), we set $k = 0.1m$ and, in Panels (b) and (c), we set $m = 200$. In all panels, we repeat each experiment 20 times.

**Results.** Figures 7(a,b) show the utility achieved by each of the decision policies and counterfactual explanations for several numbers of feature values $m$ and counterfactual explanations $k$. We find several interesting insights: (i) the decision policies given by Eq. 5 and the counterfactual explanations found by Algorithm 2 beat all other alternatives by large margins across the whole spectrum, showing that jointly optimizing the decision policy and the counterfactual explanations offer clear additional gains; (ii) the counterfactual explanations found by Algorithms 1 and 2 provide higher utility gains as the number of feature values increases and thus the search space of counterfactual explanations becomes larger; and, (iii) a small number of counterfactual explanations is enough to provide significant gains in terms of utility with respect to the optimal decision policy without counterfactual explanations.

Figure 7(c) shows the average cost individuals had to pay to change from their initial features to the feature value of the counterfactual explanation they receive. As one may have expected, the results show that, under the counterfactual explanations of minimum cost (Minimum cost and Diverse), the individuals invest less effort to change their initial features and the effort drops as the number of counterfactual explanations increases. In contrast, our methods incentivize the individuals to achieve the highest self-improvement, particularly when we jointly optimize the decision policy and the counterfactual explanations.

## F    Additional details on the experiments on real data

### F.1    Feature representation & preprocessing steps

For each applicant in the lending dataset, the label $y$ indicates whether an applicant fully pays a loan ($y = 1$) or ends up to a default/charge-off ($y = 0$) and the features $\boldsymbol{x}$ are:

- Loan Amount: The amount that the applicant initially requested.
- Employment Length: How long the applicant has been employed.
- Debt to Income Ratio: The ratio between the applicant's financial debts and her average income.
- FICO Score: The applicant's FICO score, which is a credit score based on consumer credit files. The FICO scores are in the range of 300-850 and the average of the high and low range for the FICO score of each applicant has been used for this study.
- Annual Income: The declared annual income of the applicant.

Here, we assume that all of the aforementioned features are *actionable*, meaning that an individual denied a loan can change their values in order to get a positive decision.

For each credit card holder in the credit dataset, the label indicates whether a credit card holder will default during the next month ($y = 0$) or not ($y = 1$) and the features $\boldsymbol{x}$ are:

Table 1: Dataset details

| Dataset | # of samples | Classifier | $k$ | Accuracy | $m$ | $\gamma$ |
|---------|--------------|------------|-----|----------|-----|----------|
| credit | 30000 | Logistic Regression | 100 | 80.4% | 3200 | 0.85 |
| lending | 1266817 | Logistic Regression | 400 | 89.9% | 400 | 0.97 |

- Marital status: Whether the person is married or single.
- Age Group: Group depending on the person's age (<25, 25-39, 40-59, >60).
- Education Level: The level of education the individual has acquired (1-4).
- Maximum Bill Amount Over Last 6 Months
- Maximum Payment Amount Over Last 6 Months
- Months With Zero Balance Over Last 6 Months
- Months With Low Spending Over Last 6 Months
- Months With High Spending Over Last 6 Months
- Most Recent Bill Amount
- Most Recent Payment Amount
- Total Overdue Counts
- Total Months Overdue

Here, we assume that all features except Marital Status, Age Group and Education Level are actionable and, among the actionable features, we assume that Total Overdue Counts and Total Months Overdue can only increase.

In both cases, note that the actionable features are numerical, however, our methodology only allows for discrete valued features. Therefore, rather than using the numerical values as features, we first cluster the loan applicants (or credit card holders) into $k$ groups based on the original numerical features using k-clustering and then, for each applicant (or credit card holder), use the cluster identifier it belongs to, represented using a one-hot encoding, as a feature. After this preprocessing step, the discrete feature values $x_i$ consists of all possible value combinations of discrete non-actionable features, if any, and cluster identifiers.

To approximate the values of the conditional distribution $P(y \,|\, \boldsymbol{x})$, we train four types of classifiers (Multi-layer perceptron, support vector machine, logistic regression, decision tree) using the default scikit-learn parameters and then choose the pair of classifier type and number of clusters $k$ that maximizes accuracy, estimated using 5-fold cross validation. Finally, we set $\gamma$ equal to the 50-th percentile of all the individuals' $P(y = 1 \,|\, \boldsymbol{x})$ values causing a $50\%$ acceptance rate by the optimal threshold policy in the non strategic setting. Table F.1 summarizes the resulting experimental setup for both datasets.

### F.2  Examples of counterfactual explanations

In this section, we focus on the credit dataset and look more closely into the counterfactual explanations $\mathcal{E}_m(\boldsymbol{x})$ and $\mathcal{E}(\boldsymbol{x})$ provided by the minimum cost baseline and Algorithm 1, respectively, by means of an (anecdotal) example. To this end, for a fixed $\alpha$ and $k$, we first track down the individuals whose best-response under both methods is to change their initial features to the provided counterfactual explanation. Then, for each of these individuals, we compare the counterfactual explanations provided by each of both methods.

Table F.2 shows the initial features $\boldsymbol{x}$ together with the counterfactual explanations $\mathcal{E}_m(\boldsymbol{x})$ and $\mathcal{E}(\boldsymbol{x})$ for one of the above individuals picked at random. In this example, the individual is a university student, unmarried and under the age of 25 who is advised to follow the counterfactual explanations to maintain her credit. Since the marital status, age group and level of education are all non-actionable features, both counterfactual explanations maintain the initial values for those features. Under the minimum cost baseline, the bank would advise the individual to reduce her monthly credit card bill by ~$150 and limit high spending to 2 months per semester so that her risk of default would decrease from $16\%$ to $13\%$. However, under Algorithm 1, the bank would advise to reduce her

Table 2: Counterfactual explanations $\mathcal{E}_m(\boldsymbol{x})$ and $\mathcal{E}(\boldsymbol{x})$ provided by the minimum cost baseline and Algorithm 1, respectively, to an individual with initial feature value $\boldsymbol{x}$. Initially, the individual's outcome is $P(y = 1 \,|\, \boldsymbol{x}) = 0.84$ and, after best-response, her outcome is $P(y = 1 \,|\, \mathcal{E}_m(\boldsymbol{x})) = 0.87$ and $P(y = 1 \,|\, \mathcal{E}(\boldsymbol{x})) = 0.89$, respectively. In both methods, we set $\alpha = 2$ and $k = 160$.

| Feature | $\boldsymbol{x}$ | $\mathcal{E}_m(\boldsymbol{x})$ | $\mathcal{E}(\boldsymbol{x})$ |
|---|---|---|---|
| Married | No | No | No |
| Age group | Under 25 | Under 25 | Under 25 |
| Education | Student | Student | Student |
| Maximum Bill Amount Over Last 6 Months | $2246 | $2084 | $1929 |
| Maximum Payment Amount Over Last 6 Months | $191 | $188 | $221 |
| Months With Zero Balance Over Last 6 Months | 0 | 0 | 0 |
| Months With Low Spending Over Last 6 Months | 0 | 0 | 0 |
| Months With High Spending Over Last 6 Months | 4 | 2 | 1 |
| Most Recent Bill Amount | $2145 | $2003 | $1750 |
| Most Recent Payment Amount | $123 | $124 | $100 |
| Total Overdue Counts | 0 | 0 | 0 |
| Total Months Overdue | 0 | 0 | 0 |

monthly credit card bill by ∼$400, limit high spending to 1 month per semester, and additionally increase her monthly credit card payoff slightly so that her risk of default would decrease to 11%. Since by construction, both $\mathcal{E}_m(\boldsymbol{x})$ and $\mathcal{E}(\boldsymbol{x})$ are inside the region of adaptation of $\boldsymbol{x}$, the individual is guaranteed to follow the advice in both cases, however, under Algorithm 1, the individual would be less likely to default and achieve a superior long-term well being.