[Reviews · NeurIPS 2020]

Review 1

Summary and Contributions: The authors design a variety of algorithms for finding decision policies and counterfactual explanations that maximize utility for strategic behavior.

Strengths: The paper offers a novel contribution in connecting strategic behavior and ML interpretability. The authors design a variety of algorithms for an individual's strategic behavior given the counterfactual explanations they receive from an ML system. This is a novel contribution.

Weaknesses: There are some concerns with the presentation of the problem. The paper makes a distinction between "beneficial decision" and "positive prediction" without clarifying how these two differ. These two notions run through the paper and so it is important for the reader to have a proper understanding of how do they differ. The proposal also runs the danger of using counterfactual explanations to manipulate to receivers of explanations. The authors need to think about considering some mechanisms to counter the manipulation. If not, this proposal might have some unintended ethical implications.

Correctness: Some clarifications are required as mentioned above. Although the paper is theoretically correct, I am a bit unsure what happens to the correctness of the statements when the clarifications between the two notions of "beneficial decision" and "positive prediction"are provided.

Clarity: The paper's writing is good. However, there are several (very) long setneces that make reading the paper difficult (e.g., the length of the first sentence of Section 7 is over 3 lines! I highly recommend a proper grammatical check and proof reading.

Relation to Prior Work: Yes. Although I suggest the authors add a few more sentences to contextualize their work more properly in relation to the existing literature.

Reproducibility: Yes

Additional Feedback:


Review 2

Summary and Contributions: This paper proposes and analyzes a model of strategic behavior under counterfactual explanations. In this model, a decision-maker chooses a policy and a small set of explanations that can be provided to decisions subjects who receive unfavorable decisions. In response, decision subjects follow the given explanation to improve their future outcomes. The authors show that in general, doing this optimally is NP-hard; however, they show that the problem is submodular, allowing for good approximations. They also show that certain matroid constraints allow the small set of explanations to work across multiple segments of the population simultaneously. Finally, they provide experiments using both real and synthetic data. Edit: Thanks to the authors for their response. If accepted, I'd encourage them to include a discussion of the impact on users.

Strengths: The problem being tackled here is a natural one, addressing the utility of counterfactual explanations. The analysis is fairly thorough, considering a few settings that build on one another. The experiments demonstrate that the proposed algorithms do improve the decision-maker's utility under this model.

Weaknesses: The paper's biggest omission is that it only considers decision-maker utility as opposed to social welfare/decision subjects' utility. This is significant because the model and techniques proposed are inherently extractive in the following sense: the decision-maker can and will induce the subject to pay a cost of (say) .5 in order to improve the decision-maker's utility by .01. As noted in the paper, the hope is that the improvement is worth it to both the decision-maker and the subject, but there's no guarantee that this will actually be the case. I think the experiments should at least investigate this question: does social welfare ultimately increase? Are there individuals whose utility decreases compared to the non-strategic setting? In particular, it would be interesting to see a comparison to a world where subjects have access to P(y | x) and choose the level of risk (possibly still subject to the threshold gamma) that optimizes their own utility. Beyond this, as noted by the authors, many of the assumptions made are quite strong: - All features are causal -- much of the work on strategic classification actually assumes the opposite (i.e., all strategic behavior is gaming), though it's not clear that this is any more reasonable. - The decision-maker knows the true probabilistic relationship between features and outcomes. - The number of possible feature vectors is small.

Correctness: The claims and proofs appear to be correct, though I did not thoroughly check the appendix.

Clarity: The paper is well-written for the most part. Some of the notation can be a bit tedious.

Relation to Prior Work: Yes, this work follows naturally from prior work, which is adequately discussed.

Reproducibility: Yes

Additional Feedback: - alpha, Algorithm 1, and Algorithm 2 are referenced in the main text but only defined in the appendix. - Because the cost function is key to the experiments, I think it should be discussed in the main text. - For clarity, I think pi should be defined as deterministic, since allowing it to be continuous doesn't really add much to the results. - The statement of Proposition 4 seems to allow for pi(x) = 1 for x such that P(y | x) < gamma, though the proof correctly rules this out. - The Broader Impacts section could use more discussion of how the techniques proposed here have the potential to favor the decision-maker at the expense of the decision subjects.


Review 3

Summary and Contributions: The paper considers the problem of finding a (binary) decision policy, along with a set of recommendations - more precisely, a set of examples that would have lead to a positive decision. The first part considers the problem of finding a set of recommendations for a given policy, showing that it is NP-Hard, and that the solution can be 0.632-approximated using a greedy algorithm. The second part considers the problem of simultaneously finding a good policy and a set of recommendations. Here again it is shown that the problem is NP-hard, but that finding a policy given the explanations is only polynomial. This again leads to an approximation algorithm. The complexities of both algorithms are O(k m^2), where k is the number of recommendations, and m is the number of feature values. The algorithms are then tested on two real datasets (loan and credit card data), showing that they perform well.

Strengths: The problem addressed by this paper is very relevant given current legal demands for transparency. I can see these algorithms being useful in a real-world setting. It is reasonably clear: at least the problem description and the methods are well formalized. I thought that recognizing the problem as submodular was insightful and non-trivial (the NP-hardness was less surprising). The experimental section was interesting (especially the leakage property in Fig 2).

Weaknesses: Parts of the paper weren't clear to me. For instance, to mention only a constrained subset of things I didn't understand: a) What is the “strategic setting”, mathematically or procedurally? b) What is (technically) counterfactual about the recommendations (I couldn't map it onto a causal framework)? c) What is alpha in the experimental part (is it related to gamma)? d) What were the actual proposed algorithms? (or: please describe them in the main text) e) Is the problem formulation new? In general, I thought that the text could have been gentler (and less dense with mathematical super- and subscripts).

Correctness: I did not read the appendix, therefore I cannot judge the correctness of the algorithms - it would have been nice to have them sketched out in the main paper. From the top of my head the 0.632-approximation makes sense. I can't judge the randomized algorithm.

Clarity: The paper is a tad dense but in general (see questions above) precise and correct.

Relation to Prior Work: It wasn't 100% clear to me whether the problem formulation is new. Are references [14], [15], [36], [37], and [39], which you use in your comparisons, the only ones that deal with a related problem? Otherwise the background seems well-researched.

Reproducibility: Yes

Additional Feedback: POST REBUTTAL: I'd like to thank the authors for their clarifications, and, if accepted, I'd strongly encourage to include the suggested changes into the main text.

[Author Response · NeurIPS 2020]

We would like to thank the reviewers for their comments and suggestions. We will incorporate their feedback in the revised version of the paper.

**Reviewers #1 and #2.** As discussed in lines 125-135, our algorithms will look for the counterfactual explanations $\mathcal{A}$ and decision policies $\pi(\mathbf{x})$ that maximize the decision maker's utility rather than the individuals' best interest[1]. As a consequence, it is true that, compared to the non-strategic setting, the policies that are optimal in the strategic setting may induce some of the subjects to pay an additional (immediate) cost to change features in order to receive a positive decision, as shown in Figure 6c in Appendix E. However, we would like to point out that any subject who would have received a positive decision under the decision policy that is optimal in the non-strategic setting will still receive a positive decision under the decision policy that is optimal in the strategic setting after they best respond. Moreover, subjects who do pay an additional cost to change features will always increase their outcomes $P(y \mid \mathbf{x})$, as shown in Figure 7 in Appendix F.3, and this is likely to increase their individual utility in the long term. In the revised version of the paper, we will expand our discussion regarding the potential of our algorithms to favor the decision-maker at the expense of the decision subjects, in light of the results shown in Figures 6c and 7.

**Reviewer #2 and #3.** If our submission is accepted, we will make use of the ninth content page of the camera-ready version to bring the definition of $\alpha$, the algorithmic boxes for Algorithms 1 and 2, and the discussion of the cost function estimation to the main text.

**Reviewer #1.** Under our problem formulation, a decision $d$ is beneficial to the individuals who are subject to (semi)-automated decision making if $d = 1$ (e.g., an individual receives a loan) and a prediction $\hat{y}$ made by a machine learning model is positive if $\hat{y} = 1$ (e.g., an individual repays a loan). In this context, note that, rather than explaining predictions by machine learning models as in previous work, we pursue the development of methods to find counterfactual explanations for the decisions, as argued in lines 38-42. We will clarify this in the revised version of the paper.

We will expand our comparison with the existing literature and further discuss the necessity to distinguish between decisions and predictions, as argued by several authors in a series of recent papers [23, 25-27, 47, 48].

**Reviewer #2.** As noted by the reviewer, some of our assumptions are quite strong, however, we still think they do not nullify our contributions, especially given the paucity of work in the area. That being said, we are hopeful to relax some of these assumptions in future work.

We will fix the statement of Proposition 4.

**Reviewer #3.** The "strategic setting" refers to a scenario in which individuals who are subject to (semi)-automated decision making use knowledge, gained by explainability, to change their own features to maximize their chances of receiving a beneficial decision. In our work, we formally characterize this setting mathematically for counterfactual explanations.

A counterfactual is a statement of how the world would have to be different for a desirable outcome to occur [13]. In our problem formulation, the world are the feature values $\mathbf{x}$, the desirable outcome is the positive decision $d = 1$, and the statement is the counterfactual explanation $\mathcal{E}(\mathbf{x})$. Given an individual with initial feature values $\mathbf{x}$ who would receive a negative decision $d = 0$, the counterfactual explanation provides her with an example of a feature value $\mathcal{E}(\mathbf{x})$ under which she is guaranteed to receive a positive decision $d = 1$. We will clarify this in the revised version of the paper.

The problem formulation is new. Previous work on counterfactual explanations [13-15, 36-37, 39] has focused on explaining predictions, rather than decisions, and has not investigated the connection between strategic machine learning and explanations. The most closely related work is by Tabibian et al. [23] in the strategic machine learning literature, however, they have considered a setting where decision makers share their entire policies with the individuals subjects to their decisions rather than counterfactual explanations. In this context, please note that we have included further related work in Appendix A. If our submission is accepted, we will make use of the ninth content page of the camera-ready version to bring that content to the main.

## Footnotes

[1]We did not explicitly use the wording social welfare or decision subject's utility, however, it is in the individuals' best interest to maximize their utility.


[Meta-Review · NeurIPS 2020]

This paper proposes and analyzes a model of strategic behavior under counterfactual explanations. In this model, a decision-maker chooses a policy and a small set of explanations that can be provided to decisions subjects who receive unfavorable decisions. In response, decision subjects follow the given explanation to improve their future outcomes. While doing so is NP Hard, the resulting formulation is shown to be submodular allowing for efficient approximations. This paper establishes an interesting connection between strategic behavior and explainability. While the paper makes some interesting contributions, it has the following weaknesses: 1) Certain notions are not well defined -- "beneficial decision" vs. "positive prediction" 2) Actual algorithms pushed to appendix 3) Strong assumptions are being made -- e.g., only considering decision-maker utility as opposed to social welfare/decision subjects' utility; assuming decision-maker knows the true probabilistic relationship between features and outcomes. We would encourage the authors to address the aforementioned aspects in their final version.